# Comparative Studies of Bioactivities and Chemical Components in Fresh and Black Garlics

**DOI:** 10.3390/molecules29102258

**Published:** 2024-05-11

**Authors:** Kanako Matsuse, Sho Hirata, Mostafa Abdelrahman, Tetsuya Nakajima, Yoshihito Iuchi, Satoshi Kambayashi, Masaru Okuda, Kimiko Kazumura, Benya Manochai, Masayoshi Shigyo

**Affiliations:** 1Division of Yamaguchi University and Kasetsart University Joint Master’s Degree Program in Agricultural and Life Sciences, Yamaguchi University, 1677-1 Yoshida, Yamaguchi 753-8515, Japan; 2Laboratory of Agroecology, Department of Bioresource Sciences, Faculty of Agriculture, Kyushu University, Motooka, Nishi-ku, Fukuoka 819-0395, Japan; hirata.sho.481@m.kyushu-u.ac.jp; 3Institute of Genomics for Crop Abiotic Stress Tolerance, 1006 Canton Ave, Lubbock, TX 79409, USA; 4Graduate School of Sciences and Technology for Innovation, Yamaguchi University, 1677-1 Yoshida, Yamaguchi 753-8515, Japanyiuchi@yamaguchi-u.ac.jp (Y.I.); 5Laboratory of Veterinary Internal Medicine, Joint Faculty of Veterinary Medicine, Yamaguchi University, 1677-1 Yoshida, Yamaguchi 753-8515, Japan; s-kam@yamaguchi-u.ac.jp (S.K.); okudamu@yamaguchi-u.ac.jp (M.O.); 6Global Strategic Challenge Center, Hamamatsu Photonics K.K., 5000 Hirakuchi, Hamana-ku, Hamamatsu 434-8601, Japan; kimiko.kazumura@hpk.co.jp; 7Department of Horticulture, Faculty of Agriculture, Kasetsart University, Bangkok 10900, Thailand; benya.m@ku.ac.th

**Keywords:** *Allium sativum*, black garlic, functional food, anticancer activity, processed food

## Abstract

To investigate the bioactivities of fresh garlic and its processed product, black garlic, we conducted comparative analyses of antioxidant, anti-inflammatory, innate immune activation, and anti-cancer activities in addition to the chemical composition (sugar, amino acid, and polyphenol contents) of these materials. Simultaneous assay using neutrophil-like cells showed that fresh garlic exhibited antioxidant and innate immunostimulatory activities, whereas black garlic displayed a potent anti-inflammatory effect. The antioxidant activity index was correlated with phenol and flavonoid contents, while the innate immunostimulatory activity was correlated with fructan content. Furthermore, some black garlics with low fructose content were found to inhibit the proliferation of UM-UC-3 cancer cells, while other black garlics rich in fructose increased UM-UC-3 cell proliferation. It was shown that the processing of fresh garlic could change the composition of sugars, antioxidants, and amino acids, which have different effects on neutrophil-like cells and UM-UC-3 cells, as well as on bioactivities.

## 1. Introduction

Garlic (*Allium sativum* L.) bulb is believed to contain various bioactive substances that contribute to health, such as organosulfur compounds, phenolic compounds, and polysaccharides [1], and it is used worldwide as a spice and ingredient in other foods and pharmaceuticals. Garlic has been cultivated and used since ancient times; it was already being cultivated and eaten in ancient Egypt [2]. Garlic is also known to produce a variety of bioactive components and exhibit different bioactivities depending on the method of preparation or processing. Black garlic is a processed food produced by aging fresh garlic for 10 to 40 days at 60 to 90 °C under a humidity-controlled environment of 70 to 90% [3]. Black garlic is characterized by dark brown turned scales, which is due to the Maillard reaction [4]. Black garlic has no pungent smell peculiar to garlic and has a sweet taste like dried fruit. Furthermore, black garlic has been reported to exhibit a number of bioactivities, including antioxidant, anti-inflammatory, and anti-cancer effects [5,6,7]. Sulfides, cysteine derivatives, allicin, and other sulfur-containing compounds have been proposed as key factors in garlic’s antioxidant and other bioactivities [8,9,10]. On the other hand, non-sulfur compounds commonly found in plants, such as polyphenols, are also known to have health-promoting effects [11,12]. The biological properties of garlic may be due to the synergistic effects of various phytochemicals in garlic and their proportions. However, few studies have conducted comprehensive comparative analyses of the differences in bioactivities and content composition among the varieties of garlic used to produce black garlic. Therefore, there are still many unclear points regarding changes in the chemical compositions during the production process of black garlic and the relationship between the various biological activities and active compounds. As indices for evaluating the bioactivity of foods, many in vitro experimental systems, such as radical scavenging and chemiluminescence, are used in experiments to evaluate antioxidant effects, and many experimental systems using model experimental animals are used to evaluate anti-inflammatory effects. However, these systems do not accurately reflect in vivo effects and have cost and ethical problems. Therefore, the Simultaneous Evaluation Cell Assay for Antioxidant, Anti-inflammatory, and Innate Immune Activation was developed by Kazumura et al., which utilizes the biological defense response of innate immune cells to simultaneously measure not only the reactive oxygen species (ROS) scavenging capacity of foods, but also the effects of foods on innate immune function [13]. In this study, the bioactivities of fresh and black garlic were evaluated by four methods: (1) evaluation of antioxidant, anti-inflammatory, and innate immune activation effects using the Simultaneous Evaluation Cell Assay, (2) measurement of 1,1-Diphenyl-2-picrylhydrazyl (DPPH) radical scavenging activity, (3) measurement of the viability of cancer cells, and (4) evaluation of total phenol, flavonoid compounds, carbohydrate, and amino acids contents.

## 2. Results and Discussion

### 2.1. Simultaneous Monitoring of Superoxide and Intracellular Calcium Ions in Neutrophils by Chemiluminescence and Fluorescence

Simultaneous monitoring of superoxide and calcium ion concentrations in neutrophil-like cells stimulated with chemotactic peptide f-MLP by chemiluminescence and fluorescence, respectively, was used to evaluate bioactivities [13]. The bioactivities evaluated in this experiment were antioxidant, anti-inflammatory, and innate immune activation effects. Compared to the control, if calcium ions were taken up normally and superoxide (O_2_^−·^) was scavenged by the sample, it was determined to be an antioxidant effect. If calcium ion uptake into the cells was inhibited and superoxide production was suppressed, it was determined to be anti-inflammatory. When calcium ion uptake was similar to or higher than that of the control and superoxide production was higher, it was determined to be an innate immune activation effect. In fresh garlic, with the exception of ‘White-roppen’ (obtained in 2019), the calcium ion influx was kept stable (Figure 1A) and the concentration of O_2_^−·^ was significantly reduced (Figure 1B) as compared to the control at extract concentrations of 0.3 to 3.0 × 10^−2^ mg mL^−1^. Thus, these garlics were confirmed to exhibit antioxidant activity at 0.3 to 3.0 × 10^−2^ mg mL^−1^ (Figure 1A,B; Appendix A). Fresh garlic ‘White-roppen’ (obtained in 2019) showed antioxidant activity at concentrations ranging from 1.0 × 10^−3^ to 3.0 × 10^−3^ mg mL^−1^ (Figure 1A,B; Appendix A). In addition, fresh garlic clones ‘Iki-ooninnniku’, ‘Chile-60’, and ‘Spain-225’ showed no change in calcium ion influx at 1.0 mg mL^−1^, and the intensity of O_2_^−·^ increased significantly. Therefore, these 1.0 mg mL^−1^ extracts of fresh garlic were suggested to have an innate immunostimulant effect (Figure 1A; Appendix A). A high positive correlation (*r* = 0.93) was observed between the fructan content and neutrophil-like cell O_2_^−·^ production (Figure 2A). The higher content of fructan in garlic tended to increase O_2_^−·^ production by neutrophil-like cells, corroborating that O_2_^−·^ production is related to innate immune responses and that fructan content is strongly associated with the innate immune activation effects of neutrophil-like cells. Polysaccharides derived from plant materials have immunostimulatory properties [14] and also have stimulating effects on neutrophils, which are responsible for innate immunity. Fresh garlic showed antioxidant and innate immune activation effects, while black garlic showed anti-inflammatory effects. In all black garlic extracts, the influx of calcium ions and the intensity of O_2_^−·^ were significantly reduced, and anti-inflammatory effects were observed when high concentration extracts of 0.1 to 1.0 mg mL^−1^ were added (Figure 1A,B; Appendix A). In particular, black garlic purchased in Indonesia showed a strong anti-inflammatory effect, with a significant decrease (*p* < 0.05) in reactive oxygen species concentration and intracellular calcium ion concentration at the high concentration. Furthermore, the same garlic showed different bioactivities before and after processing. Such differences are crucial in the context of targeted therapeutic applications, where fresh garlic’s properties may be better suited for enhancing immune function, whereas black garlic’s enriched bioactive compounds make it a potential candidate for anti-inflammatory and cancer-preventative strategies.

### 2.2. Evaluation of Antioxidant Activity

DPPH radical scavenging activity and total phenol and flavonoid content, which are antioxidant components, were significantly higher in black garlic than in fresh garlic. Total phenolic and flavonoid compound content showed a high negative correlation with O_2_^−·^ production and a high positive correlation with DPPH radical scavenging activity (Figure 3). In other words, the higher the total phenolic and flavonoid content, the stronger the antioxidant capacity to scavenge O_2_^−·^ and DPPH radicals. The results suggest that the increase in antioxidant capacity due to the processing of black garlic is due to the increase in total phenolic and total flavonoid compounds [15]. Many phenolic and flavonoid compounds in plants are found to be glycosides [16]. In some reports, heating plants inhibits the activities of oxidative and hydrolytic enzymes that destroy antioxidant components, whereas tissue disruption raises the available antioxidant content and increases antioxidant capacity [17]. Furthermore, in an experiment to evaluate the bioactivities of anthocyanidin and anthocyanin using the innate immune response, anthocyanidins such as cyanidin and delphinidin showed antioxidant and anti-inflammatory effects, while anthocyanins, such as cyanidin 3-glucoside, cyanidin 3-rutinoside, delphinidin 3-glucoside, and delphinidin 3-rutinoside, showed innate immune activation [18]. Therefore, it is assumed that heat treatment during the processing of black garlic might increase available phenolic compounds, leading to high antioxidant activity in black garlic. These findings align with our hypothesis that processing garlic transforms its biochemical makeup, thereby enhancing its functional properties and expanding its applicability in health and disease prevention strategies. This comprehensive evaluation of bioactivities facilitates a deeper understanding of how specific compounds contribute to the health-promoting effects of garlic, guiding more targeted use in nutritional and therapeutic applications.

It is worth noting that the antioxidant activity in neutrophil-like cells involves more than just neutralization of free radicals as in the DPPH assay. It includes modulation of cellular oxidative stress responses, which can be influenced by a broader range of compounds present in fresh garlic, not solely phenolics and flavonoids. Fresh garlic contains other antioxidant compounds that may not directly contribute to DPPH scavenging but are effective in cellular systems by modulating oxidative stress pathways and enhancing cellular antioxidant defenses. Assay Specificity: The results highlight the importance of assay choice in evaluating the antioxidant capacity of natural products. Cellular assays can provide insights into the biological relevance of antioxidant activity, which chemical assays like DPPH might not fully capture [19].

### 2.3. Evaluation of Anticancer Effect

Cancer cell viability was reduced in ‘Black Garlic M’ and ‘Black Garlic’ when the concentration of the sample extracts in the medium was as high as 0.3–1.0 mg mL^−1^. On the other hand, ‘White-roppen’ and its processed black garlic increased the activity of cancer cells and stimulated their proliferation when the concentration of the extract was increased (Figure 4; Appendix A). Cancer is a disease caused by genetic errors. Most reactive oxygen species generated in cells are removed by scavenging enzymes and antioxidants, but when overproduction occurs or the scavenging system is unable to keep up, oxidative DNA damage increases and the possibility of genetic problems that lead to carcinogenesis is high [20]. This suggests that black garlic, which exhibits radical scavenging and antioxidant activity, may contribute to the suppression of cancer development. The inhibitory effect on cancer cell proliferation was examined in terms of the induction of apoptosis. S-allylmercaptocysteine inhibits the proliferation of two leukemia-derived cell lines in a concentration-dependent manner [21]. It has been suggested that ajoene may induce apoptosis by increasing the production of peroxides and activating transcription factors [22]. The results indicate that the high radical scavenging and antioxidant effects of black garlic contributed to the inhibition of cancer cell viability, and the apoptosis-inducing substances inhibited the proliferation of cancer cells. There was a high positive correlation between the fructose content of black garlic and the growth rate of cancer cells (Figure 2B; Appendix A). When testing the addition of fructose to UM-UC-3 cells, the activity of UM-UC-3 cells increased with increasing concentrations of fructose in the medium and was significantly higher at 250 µg mL^−1^ as compared to the control (Figure 5; Appendix A). Cancer cells generally require large amounts of energy and are believed to use the glycolytic pathway, which provides energy quickly, as their main pathway of energy acquisition [23]. In this case, the large amount of fructose contained in black garlic extract may have provided the energy source for cell proliferation. It has also been reported that the addition of fructose to pancreatic cancer-derived cell lines stimulated the synthesis of nucleic acids and nucleotides and increased proliferation [23]. It is possible that the same activation of the nucleic acid synthesis pathway in UM-UC-3 cells also led to cell proliferation. These results suggest that fructose in the medium increased the activity of UM-UC-3 cells and that the effect of fructose in black garlic on the proliferation of UM-UC-3 cells exceeded the effect of anti-cancer substances when the fructose content in black garlic was increased.

### 2.4. Amino Acid and Sugar Contents

Amino acid content was measured, and a total of 34 amino acids were found. The amino acids in fresh and black garlic are quantitatively and qualitatively different. Fresh garlic tended to have higher total amino acid content than black garlic, with arginine and asparagine accounting for most of the total amino acid content. On the other hand, black garlic tended to have lower total amino acid content than fresh garlic but higher contents of amino acids such as alanine, glycine, tyrosine, and phosphoserine than fresh garlic (Figure 6; Appendix A). The Maillard reaction occurs in the process of black garlic production. In their study of the antioxidant components of aged garlic extract, Ryu et al. isolated fructosyl arginine, a Maillard compound that exhibits strong hydrogen peroxide scavenging activity [24]. The formation of this compound correlates with the formation of glucose in garlic extract during ripening and is considered to be formed by the non-enzymatic reaction of fructose—produced by the breakdown of fructan—with arginine, which is present in large quantities in garlic [25]. It is possible that during the black garlic production process, amino acids were consumed in the same way as in the aged garlic extract, resulting in the formation of Maillard compounds, which have antioxidant properties, thus increasing the antioxidant activity of the garlic. Regarding carbohydrate content, fructan was the major sugar in fresh garlic and fructose in black garlic. Fructan content decreased after processing into black garlic, and conversely, the fructose and glucose contents increased (Figure 7; Appendix A). This supports previously reported results [26]. During the heat treatment process, fructan, the main storage sugar of garlic, was decomposed and fructose and glucose accumulated. The results suggest that the dried fruit-like sweetness and stickiness characteristics of black garlic are produced in this process.

This study comprehensively evaluates the bioactive effects and biochemical compositions of fresh and black garlic, revealing distinct therapeutic potentials influenced by their specific constituents. Fresh garlic, rich in fructan, consistently exhibited antioxidant and immune-stimulating activities, marked by a stable calcium influx and reduced superoxide production, aligning with its role in enhancing innate immune responses. In contrast, black garlic, processed through thermal methods, demonstrated significant anti-inflammatory properties and higher DPPH radical scavenging abilities due to increased phenolic and flavonoid contents. Moreover, the thermal processing altered the sugar composition from fructan to higher levels of fructose and glucose, contributing to the distinct bioactivities observed, including a potent inhibition of cancer cell proliferation at certain concentrations. Interestingly, the increased fructose content in black garlic was associated with enhanced cancer cell viability, underscoring the complexity of its impact on cellular metabolism. Additionally, amino acid profiling revealed that black garlic contained lower overall amino acids but higher levels of specific compounds like alanine and glycine, likely due to the transformation involved in the Maillard reaction during its processing. These findings collectively highlight how the intrinsic properties of garlic can be modulated by cultivation and processing techniques, significantly affecting their functional applications in health and disease management, demonstrating a need for careful consideration of garlic type and preparation method in both dietary and therapeutic contexts.

## 3. Materials and Methods

### 3.1. Plant Materials

Fresh and black garlic were collected from Japan and abroad. In addition, three garlic accessions managed by Yamaguchi University were used in this study (Table 1). These accessions were obtained from local markets or national institutions in each country. Detailed information regarding these accessions was reported by Etoh [27] and Hirata et al. [28]. ‘White-roppen’ black garlic was processed in a rice cooker using warm mode for two weeks. The freeze-dried powder of fresh and black garlic bulbs was used as the experimental material. To prevent the loss of components, they were cut into thin slices on ice, immersed in liquid nitrogen for a few seconds, and then dried in a freeze-dryer (TAITEC vacuum freeze-drying equipment VD-250R) for 3 weeks. For each sample, 1 to 5 bulbs were prepared (Table 1).

### 3.2. Sample Preparation

#### 3.2.1. Hot Ethanol Extraction

The freeze-dried powder of each sample was extracted with 70% ethanol to obtain the extract. Twenty mg of the freeze-dried powder was weighed and placed in a 15 mL plastic tube. Then, 2.5 mL of 70% ethanol was added to the tube, this ratio ensures sufficient solvent volume to effectively solubilize the bioactive components from the powder, maximizing the extraction process. The mixture was mixed with a vortex mixer for 5 min, which helps disrupt the cell structure of the dried material and enhances the solubility of the bioactive compounds in the ethanol. Next, the mixture was heated at 80 °C for 15 min and sonicated for 5 min to further assist in the breakdown of cellular matrices and increase the kinetic energy of the solvent molecules. Finally, the supernatant was collected after centrifugation. The extracts were stored at −25 °C until measurement.

#### 3.2.2. Dimethyl Sulfoxide (DMSO) Extraction

The experimental samples were extracted with dimethyl sulfoxide (DMSO), and the extracts were used for the measurement. DMSO was selected as the solvent due to its ability to dissolve a wide variety of compounds, including both hydrophobic and hydrophilic substances. This property makes it particularly useful for extracting diverse bioactive molecules from biological materials without causing significant damage. Ten mL of DMSO was added to 2 g of freeze-dried sample. After mixing with a vortex mixer for 10 min, the samples were centrifuged (25 °C, 300× *g*, 5 min), and the supernatant was collected. The extracts were stored at −80 °C until measurement. The sample extracts were diluted to appropriate concentrations with DMSO.

### 3.3. Simultaneous Evaluation for Antioxidant, Anti-Inflammatory, and Innate Immune Activation

The bioactivities of plant materials were evaluated according to the method of Kazumura [13] which utilizes the innate immune response of neutrophils. We utilized neutrophil-like cells obtained by inducing the differentiation of human acute promyelocytic leukemia cell line (HL-60) as surrogates for neutrophils. The HL-60 cell line was obtained from the American Type Culture Collection (ATCC). Cells were cultured in RPMI-1640 medium (FUJIFILM Wako Pure Chemical Corporation, Osaka, Japan) supplemented with 10% fetal bovine serum (FBS, Biosera, Cholet, France) and 1% Penicillin–Streptomycin (P/S, FUJIFILM Wako Pure Chemical Corporation, Japan) at 37 °C in a humidified 5% CO_2_ atmosphere. Differentiation was induced by 96 h incubation at 4.0 × 10^5^ cells mL^−1^ in medium containing 1.3% sterile DMSO. Differentiated neutrophil-like cells were incubated for 45 min in medium with a calcium ion indicator, 3 µM fluo-3 AM. Neutrophil-like cells obtained following the method above were used to prepare the assay solution. The assay solution contained 1.0 × 10^5^ cells mL^−1^ of neutrophil-like cells in RH buffer; 0.5 µM MCLA, which is a superoxide-sensitive chemiluminescence reagent; 1 mM calcium chloride as a source of calcium ions; and DMSO extraction or DMSO. A simultaneous fluorescence and chemiluminescence measurement system (CFL-C2000, Hamamatsu Photonics, Hamamatsu, Japan) was used for the assay. Data analysis was performed using dedicated analysis software that can simultaneously analyze the three data obtained from the measurements. The stimulus enhancement of the fluorescence and chemiluminescence signals was determined by automatically calculating the baseline and peak area under the curve. The peak area ratio of each sample compared to the DMSO-treated control was calculated. The *t*-test was used to determine the significant difference between the O_2_^−·^ production and calcium ion concentration of neutrophil-like cells. A *p*-value < 0.05 was considered statistically significant.

### 3.4. DPPH Radical Scavenging Activity

DPPH radical scavenging activity was measured based on the method of Kondo et al. [29]. Samples were extracted with 70% ethanol and used for assay. Extracts were used for assay immediately after extraction. Seventy-two µL of DPPH dissolved in ethanol, pH 6.0 MES (Morpholinoethanesulfonic acid monohydrate) buffer, and 30% ethanol were mixed in the 96-well plate. Then 72 µL of sample extract diluted with 70% ethanol was added. The absorbance at 520 nm was measured with a microplate reader after incubation for 20 min at room temperature in the dark. A regression equation was calculated based on the dilution ratio of the samples, and the half-inhibition concentration (IC_50_ (mgFW)) of each sample was calculated. Results were expressed as the reciprocal of IC_50_ (mgFW^−1^).

### 3.5. Measurement of Cancer Cell Viability with CCK-8 Kit

The UM-UC-3 cell line was obtained from the European Collection of Cell Cultures (ECACC). UM-UC-3 cells cultured in Dulbecco’s Modified Eagle’s Medium (DMEM) (FUJIFILM Wako Pure Chemical Corporation, Japan) supplemented with 10% FBS and 1% P/S at 37 °C in a humidified 5% CO_2_ atmosphere for 3 days were used for the assay after confirming that they had reached 70 to 90% confluence. The cells were dissociated from the flask and suspended, and the cell suspension was centrifuged (1500 rpm × 5 min) to remove the old medium. The cells were seeded into 96-well plates (200 µL, 5000 cells/well). Cells were incubated for another 24 h after adding 2 µL of DMSO or DMSO extract per well. The experiment was performed according to the protocol of the assay kit. A total of 10 µL of CCK-8 reagents (DOJINDO, Japan) were added to each well after removing 100 µL of supernatant. The treated cells were incubated at 37 °C in a humidified 5% CO_2_ atmosphere for 3.5 h. The absorbance at 450 nm was measured with a microplate reader. The cell viability was expressed as the ratio of absorbance between sample groups and the DMSO-treated control group.

### 3.6. Determination of Soluble Sugars (Glucose, Fructose, and Sucrose)

Soluble sugar content, including fructose, glucose, and sucrose, was determined by the HPLC method using hot 70% ethanol extract. Before injection, the extract was filtered through a 0.45 µm filter. The measurements were performed under the following conditions. The identification of substances was conducted by retention time. The sugar content of the sample was determined from the peak area of the standard and that of the sample.

Column: NH2 (4.0 mm inner diameter × 250.00 mm long, Kanto Chemical, Tokyo, Japan); column temperature: 40 °C; mobile phase: 80% acetonitrile solution; flow rate: 1.0 mL min^−1^; injection volume: 100 µL; detector: refractive index detector (L-2490, HITACHI); and data collection time: 20 min.

### 3.7. Determination of Fructan Content

The fructan in extracts was determined by the thiobarbituric acid method [30] with minor modification. Fructan was extracted using 70% ethanol, and the resulting extract was then diluted fivefold with 70% ethanol. The diluted extract was then transferred into a test tube in a volume of 20 µL. Then, 10 μL of 25 mM ammonium acetate buffer was added. In addition, 10 μL of invertase solution was added and left at room temperature for 5 min to degrade the sucrose in the sample. Fifty μL of distilled water and 10 μL of 10 N sodium hydroxide solution were added and heated in boiling water for 10 min to decompose the fructose in the solution. After cooling rapidly in ice water, 1 mL of thiobarbituric acid solution and 1 mL of 12 N hydrochloric acid were added and heated in boiling water for 6 min. Finally, the absorbance at 432 nm was measured with a spectrophotometer. The concentration of fructan in the extract was determined using a calibration curve derived from 1-kestose (FUJIFILM Wako Pure Chemical Corporation, Japan), and the fructan content per 100 g fresh weight of plant sample was calculated.

### 3.8. Determination of Total Phenolic Compounds

The content of total phenolic compounds in the 70% ethanol extract was determined by the Folin–Ciocalteu method [31]. The 70% ethanol was diluted 5 times with distilled water. A total of 1 mL of Folin–Ciocalteu reagent was added to 1 mL of the diluted solution in a test tube. After 3 min, 1 mL of 10% sodium carbonate solution was added. The mixture was incubated for 1 h at room temperature in the dark. The absorbance at 530 nm was measured with a spectrophotometer (U-2000, Hitachi High-Technologies Corporation, Tokyo, Japan). The total phenolic compound content was expressed as milligram catechol equivalents (mg catechol) per 100 g fresh weight.

### 3.9. Determination of Total Flavonoid Compounds

The total flavonoid content was determined by the colorimetric method using hot 70% ethanol extract. The method was adopted from the approach of Vu et al. [32]. The 70% ethanol extract of the sample and *n*-hexane were placed in a test tube at a ratio of 1:1 and separated into two layers after stirring. Chlorophyll, carotenoids, and other pigments dissolved in the hexane layer were removed. The 0.5 mL of 70% ethanol layer was diluted with the same amount of 70% ethanol in a test tube, and 2 mL of 2% aluminum chloride solution was added. After allowing the reaction to proceed for 1 h in the dark, the absorbance at 420 nm was measured. The total flavonoid compound content was calculated using a calibration curve prepared with quercetin. The results were expressed as milligram quercetin equivalents (mg quercetin) per 100 g fresh weight of plant sample.

### 3.10. Determination of Amino Acid Contents

The freeze-dried powder of the sample was extracted with pH 2.2 lithium citrate buffer solution (FUJIFILM Wako Pure Chemical Corporation, Japan). Briefly, 10 mL of lithium citrate buffer was added to 50 mg of sample powder. The solution was vigorously mixed with a vortex mixer for 5 min and ultrasonicated for 5 min. The extract was centrifuged (20 °C, 5000 rpm, 30 min), and the supernatant was collected. Before injection, the extract was filtered through a 0.20 µm filter. Samples were measured immediately after extraction. The amino acid analysis system of SHIMADZU was used to measure amino acids by the post-column derivatization method. Amino acids were analyzed using the gradient from three mobile phase types (S228-21195-95, SHIMADZU, Kyoto, Japan) and were derivatized using two OPA reagents (S228-21195-93, SHIMADZU, Japan). Before use, sodium hypochlorite was added into the OPA reagents. Mobile phases and reagents were purchased from SHIMADZU. The HPLC conditions are as follows.

Column: Shim-Pack-Amino-Li (6.0 mm inner diameter × 100 mm long, SHIMADZU, Japan); injection volume: 50 µL; detector: fluorescence detector (RF-20A, SHIMADZU); excitation wavelength: 350 nm; fluorescence wavelength: 450 nm; flow of mobile phases: 0.6 mL min^−1^; flow of OPA reagents: 0.2 mL min^−1^.

The time program was based on the Shimadzu manual. From the injection to 22 min, mobile phase A was flowed 100%, and from there the concentration of mobile phase B was increased step by step to reach 100% at 111 min. From 133 min, mobile phase C was flowed to wash the column. The analysis was carried out for 163 min.

### 3.11. Statistical Analysis

Statistical analyses were performed with EZR [33], which is a graphical user interface for R (The R Foundation for Statistical Computing, Vienna, Austria). More precisely, it is a modified version of R Commander designed to add statistical functions frequently used in biostatistics. Each sample was measured three times (HPLC analysis was performed once), and data were expressed as the mean ± SD. Except simultaneous evaluation for antioxidant, anti-inflammatory, and innate immune activation, a one-way ANOVA followed by Dunnett’s test was used to determine the difference between groups. A *p*-value < 0.05 was considered statistically significant.

## 4. Conclusions

Both fresh and black garlic offer a range of health benefits and exhibit distinct bioactivities that are influenced by the methods used to process fresh garlic into black garlic, as well as by the specific types of garlic employed. Thermal processing alters their biochemical composition, affecting sugars, antioxidants, and amino acids, impacting neutrophil-like and cancer cells differently. Fresh garlic’s higher fructan content boosts antioxidant and immune-stimulatory effects, while black garlic, rich in phenolic and flavonoid compounds, shows pronounced antioxidant and anti-inflammatory activities. This increase in bioactive compounds boosts black garlic’s DPPH radical scavenging capabilities significantly. Moreover, the observed differential impacts on cancer cell viability, particularly the variable effects on UM-UC-3 cell proliferation, underscore the distinct ways in which each garlic form influences cellular behavior. Fresh garlic enhances cellular antioxidant defenses, while black garlic modulates energy pathways in cancer cells, showcasing their diverse therapeutic potentials. These results support the idea that processing methods and biochemical profiles significantly influence garlic’s functional properties and potential health benefits, providing insights for targeted therapeutic applications in health and disease management.

## Figures and Tables

**Figure 1 molecules-29-02258-f001:**
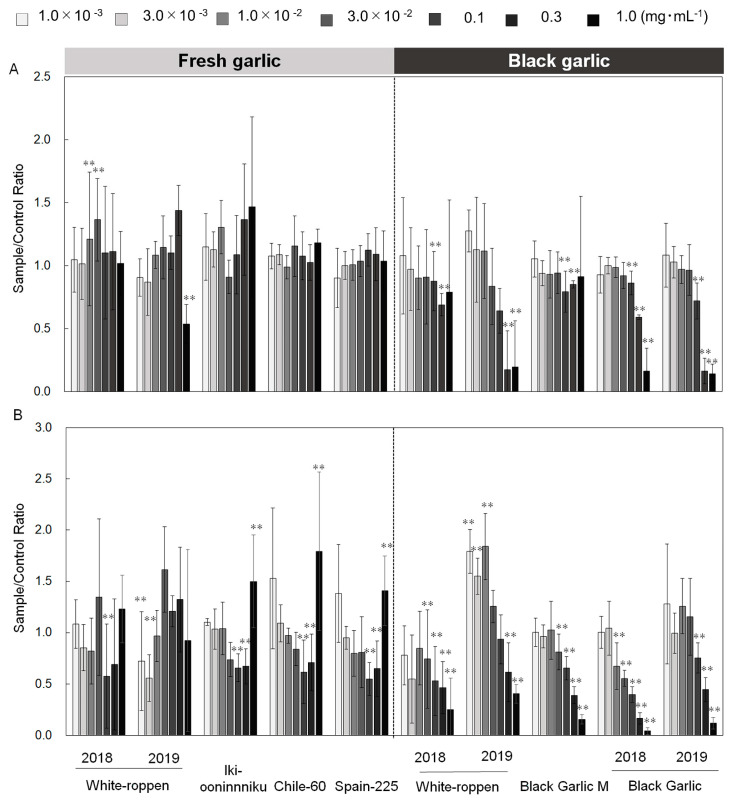
Effects of fresh and processed black garlic extracts on superoxide production and calcium ion uptake in neutrophil-like cells after exposure to various concentrations of garlic extracts ranging from 0.3 × 10^−2^ to 3.0 × 10^−2^ mg mL^−1^ for fresh garlic and from 0.1 to 1.0 mg mL^−1^ for black garlic. (**A**) Calcium ion uptake and (**B**) superoxide production by neutrophil-like cells induced by fresh and black garlic. The *y*-axis measures the relative intensity of superoxide production and calcium ion uptake compared to the DMSO-treated control. Differences between the garlic-treated cells and control are indicated by asterisks: ** (*p* < 0.01), showing statistically significant decreases in superoxide production or calcium ion uptake. The bar indicates the standard deviation (±SD) for three independent experiments (*n* = 3).

**Figure 2 molecules-29-02258-f002:**
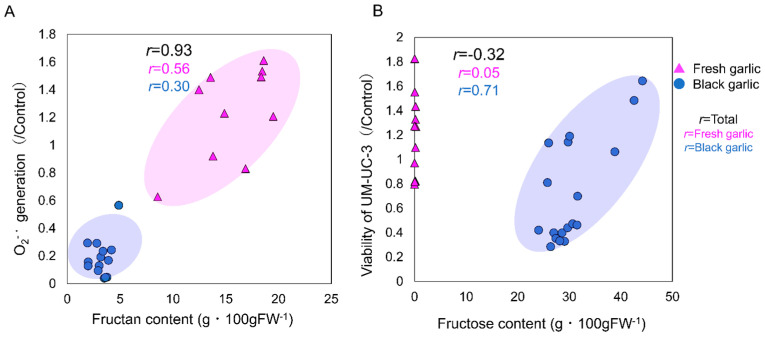
Correlation between fructan and fructose contents (g·100 FW^−1^) in fresh and black garlics and biological activities. (**A**) The relationship between the fructan content in garlic and the production of superoxide (O_2_^−·^) by neutrophil-like cells, which is an indicator of innate immunostimulatory activity. The *x*-axis represents the fructan content in fresh garlic, and the *y*-axis shows the corresponding intensity of O_2_^−^ production. (**B**) the correlation between fructose content in garlic and its effect on cancer cell activity. The *x*-axis represents the fructose content in both fresh and black garlic, and the *y*-axis measures a generic indicator of cancer cell activity. Both panels include correlation coefficients (*r*) to quantify the strength of the relationships depicted.

**Figure 3 molecules-29-02258-f003:**
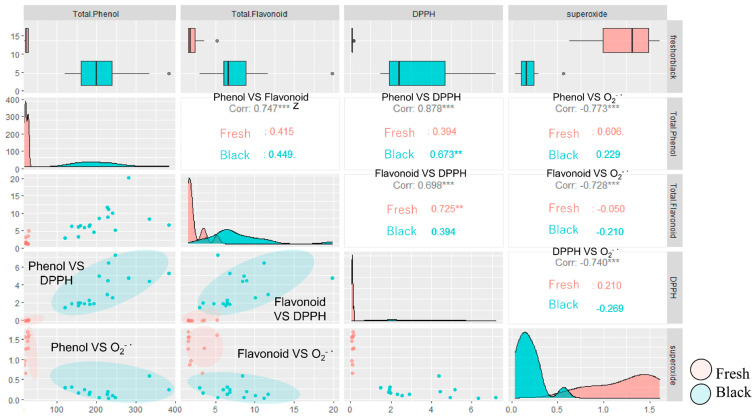
Relationship between total phenolic and flavonoid contents and their respective antioxidant activities as measured by O_2_^−·^ scavenging and DPPH radical scavenging assays. The correlation coefficients, denoted by ‘z’ symbols within the graph, measure the strength of these relationships. Statistically significant correlations between the content of these compounds in garlic and their antioxidant effects are marked by asterisks: ** (*p* < 0.01) and *** (*p* < 0.001), highlighting the robustness of the observed associations.

**Figure 4 molecules-29-02258-f004:**
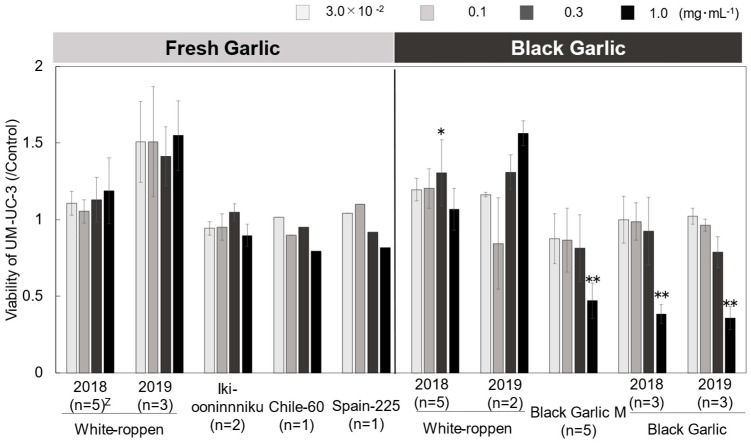
Effects of fresh and black garlic extracts on cell viability of UM-UC-3. The *x*-axis represents the concentration of garlic extracts (ranging from 0.3 to 1.0 mg mL^−1^), while the *y*-axis quantifies cell viability relative to the DMSO-treated control. Statistical significance is denoted by asterisks: * (*p* < 0.05) and ** (*p* < 0.01). Error bars represent the standard deviation (±SD) from multiple experiments (n = 1–5 biological replicates × 3 technical replicates). Z represents the number of biological replicates. Chile-60 and Spain-225 are shown in the figure as a reference because only one biological replicate was obtained for these two materials.

**Figure 5 molecules-29-02258-f005:**
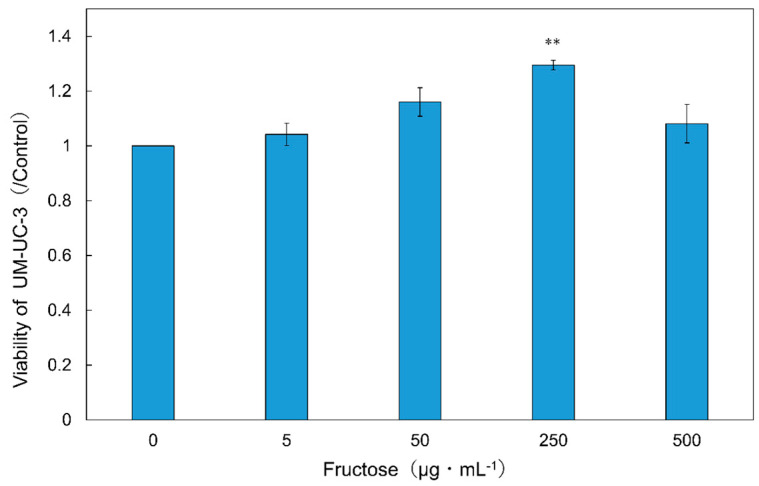
Effect of fructose on the cell viability of UM-UC-3, a bladder cancer cell. The *x*-axis shows the fructose concentration in the medium, and the *y*-axis measures cell viability as a percentage of the control (no fructose added). Data points show that fructose significantly enhances the proliferation of cancer cells, particularly at a concentration of 250 µg mL^−1^, with statistical significance indicated by ** (*p* < 0.01) as determined by Dunnett’s test. The bar represents the standard deviation (±SD) based on three replicates (*n* = 3).

**Figure 6 molecules-29-02258-f006:**
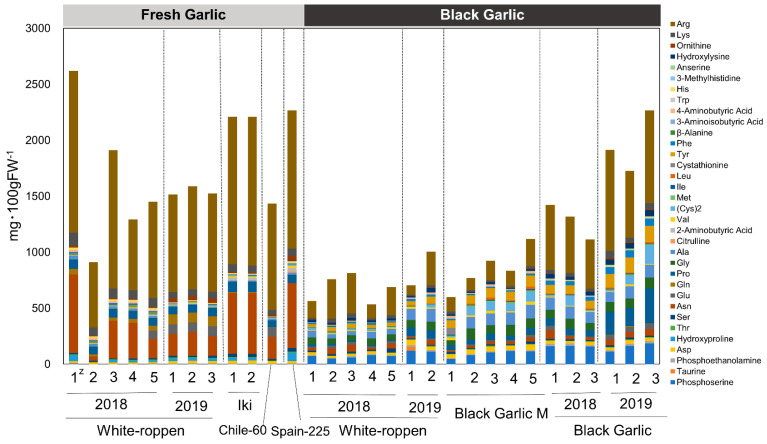
Amino acid contents of fresh and black garlic (mg 100 gFW^−1^). The *y*-axis compares the concentrations of various amino acids identified in both garlic types. The ‘z’ symbol within the graph represents the number of garlic bulbs analyzed, with each type of garlic assessed using 1–5 bulbs.

**Figure 7 molecules-29-02258-f007:**
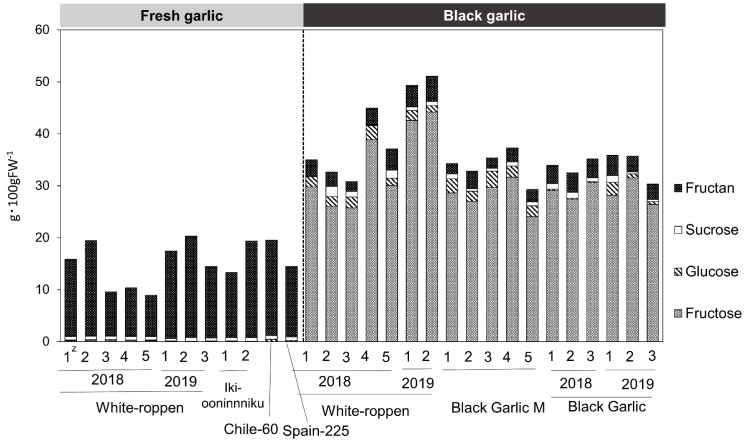
Carbohydrate contents in fresh and black garlic (g 100 gFW^−1^). The *y*-axis compares the concentrations of carbohydrates, including fructan, sucrose, glucose, and fructose, identified in both garlic types. The ‘z’ symbol within the graph represents the number of garlic bulbs analyzed, with each type of garlic assessed using 1–5 bulbs.

**Table 1 molecules-29-02258-t001:** Plant materials used in this study.

Materials		Number of Bulbs	Collected Site	Accession Information
Fresh garlic	‘White-roppen’ 2018	5	Japanese local market	—
	‘White-roppen’ 2019	3	Japanese local market	—
	‘Iki-ooninnniku’	2	Saga University, Japan	Hirata et al. [28]
	‘Chile-60’	1	Chile	Etoh [29]
	‘Spain-225’	1	Spain	Hirata et al. [28]
Black garlic	‘White-roppen’ 2018	5	Processed White-roppen 2018	—
	‘White-roppen’ 2019	2	Processed White-roppen 2019	—
	Black Garlic M	5	Japanese local market	—
	Black Garlic 2018	3	IPB Shop Official	—
	Black Garlic 2019	3	IPB Shop Official	—

## Data Availability

The original contributions presented in this study are included in the article and Appendix A; further inquiries can be directed to the corresponding authors.

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
