# Peer review of "Comparative Studies of Bioactivities and Chemical Components in Fresh and Black Garlics"

_molecules, 2024, doi:10.3390/molecules29102258_

Round 1

Reviewer 1 Report

Comments and Suggestions for Authors

Just as authors indicated in their title, they have conducted comparative studies of bioactivities and chemical components in Fresh and Black Garlics which is what they reported in the manuscript. 

The study is well written and easy to understand, however, these are my comments:

1. Abstract section last sentence - Authors when stated that "It was shown that the thermal processing steps of fresh garlic could change the composition of sugars, antioxidants, and amino acids, which have different effects on neutrophil-like cells and UM-UC-3 cells, as well as on bioactivities." Authors should revise this statement, as no where in the result and discussion section did the authors indicate that neutrophil-like cells were used to evaluate thermal processed fresh garlic. However, if authors want to retain the sentence, they should provide more data in terms of the thermal processed garlic assayed using the neutrophil-like cells.

2. Line 70-71 - "Simultaneous Evaluation Cell Assay described above," The materials and methods are now below. Authors to take note

3. Authors reported the results of the antioxidant activity for the fresh garlic using the neutrophil-like cells (Line 86 to 91) which showed activity, however for DPPH radical scavenging activity the black garlic was more active (122 to 123). Authors should give more information for the disparity on why the fresh garlic had no activity.

4. Line 98 to 99 "The higher content of fructan in garlic tended to increase O2- production by neutrophil-like cells, suggesting that O2- production is related to innate immune responses....." Authors to replace suggesting to corroborating because it is a known fact that superoxide production is related to immune response.

5. Line 108" ........with a significant decrease in", author to include the p-value after the significant decrease

6. Authors have discussed the results of their findings individually but should in one paragraph discuss the implication/relevance on the low/high bioactivities and chemical components in Fresh and Black Garlics in respect to your introduction.

7. Figure 1, Authors did not indicate where Figure 1B can be found

8.     3.2.2 Dimethyl sulfoxide (DMSO) extraction section -

Line 249 "........and only the supernatant was used." Author to revise. My suggestion "and the supernatant was collected"

Line 250 "The samples were diluted with DMSO". Author to revise. My suggestion "The samples extracts were diluted to appropriate concentrations with DMSO"

9.    Line 318, 322 "The 70 % ethanol extract was diluted 5 times" Authors to explain what it means 

10. Author should revise the conclusion. The last sentence in 390 to 392 should be removed

11. Most of the references cited are old over 70% are over 10 years

Comments on the Quality of English Language

Minor errors

Author Response

Please find the attachment. Edits are highlighted in red.

Reviewer 2 Report

Comments and Suggestions for Authors

The present Article, molecules-2893577, entitled: (Comparative Studies of Bioactivities and Chemical Components in Fresh and Black Garlics)

The Research article is contained a good topic and written in a good way. However, I have the following comments:

The article focused on investigation of the bioactivities of fresh garlic and its processed product, black garlic, a comparative analyses of antioxidant, anti-inflammatory, innate immune activation and 24 anti-cancer activities, in addition to chemical composition of products including sugar, amino acid, polyphenol contents.  I highly recommend to add explanations for all figures legends. Figure (1-7) and quality of Figures (1, 3) is very poor.  Please follow journal guidelines in quoting references inside text and in the reference list. In page (5) lines 176-179 is not relevant to this work. The rationale of extraction using dimethyl sulfoxide and hot ethanol need elaborations, I recommend to add images to confirm cytotoxicity. The conclusion section was well written according to performed experiments. It will be better to add titles and simple explanations in supplementary and unpublished materials. Most of the quoted references were old and I recommend to add updated references.

Comments on the Quality of English Language

Minor editing of English language required

Author Response

Please find attachment. Edits are highlighted in red.

Reviewer 3 Report

Comments and Suggestions for Authors

The article 'Comparative Studies of Bioactivities and Chemical Components in Fresh and Black Garlics' by Matsuse et al. at first glance appears to be an interesting and valuable scientific report, but after a thorough revision of the text, many doubts arise.

I do not really understand what the novelty of the article is based on. The analysis of the properties of garlic, especially black garlic, is a very topical subject at the moment, many papers have now been written on this topic.

The authors have evaluated the anti-cancer, anti-inflammatory activity, or the sugars and acid content of fresh and black garlic, but has no one done this before?  The discussion explains very well the mechanisms responsible for the processes involved, or the relationship of the different parameters, but I miss the reference to other work on garlic. How did the anticancer/antioxidant/anti-inflammatory activity of garlic compare to previous work. The authors did not relate their results to previously published reports. Hence, my doubts about the novelty of the manuscript.

Furthermore, the cited literature is very out of date. Many of the items cited are not within the last 20 years, let alone the last 5 years. I ask for a thorough revision of this part of the paper, especially the literature relating to the introduction and the discussion of the results.

What I also miss is the provision of the counter values that were obtained in the individual tests. For example, from Fugure 3 it is hard to deduce how many polyphenols or flavonoids there actually are in fresh and black garlic, and most importantly what the difference between the two is (from the text it is also hard to deduce this). Because from what I can guess this is the purpose of this work.

What is missing in the cellular assays is the administration of a control for DMSO. DMSO alone has a cytotoxic effect on cells, hence the obtained results of anticancer activity for plant extracts prepared/dissolved in DMSO should be corrected for the cytotoxicity of this solvent.

Figure 4: The description states that replicates were performed "n=1-5". This seems highly disconcerting to build results, conclusions, on the basis of a single repetition, even more so in cellular assays where, due to the low reproducibility of the results obtained, the number of repetitions should reach at least 10-12.

Author Response

(The authors gave the same response as above.)

Round 2

Reviewer 1 Report

Comments and Suggestions for Authors

There are two comments I would like to make

1.  Results and discussion section from line 87, authors should present their figures in alphabetical order 1A should come before 1B

2. Authors should revise the conclusion as it is long with repetition of aspects of results and discussion.  See below, a suggested revision of the conclusion presented in the manuscript, authors are free to use it or make their own brief and informative.

Fresh garlic and black garlic both offer health benefits with distinct bioactivities influenced by processing methods and garlic types. Thermal processing alters their biochemical composition, affecting sugars, antioxidants, and amino acids, impacting neutrophil-like and cancer cells differently. Fresh garlic's higher fructan content boosts antioxidant and immune-stimulatory effects, while black garlic, rich in phenolic and flavonoid compounds, shows pronounced antioxidant and anti-inflammatory activities. Black garlic's DPPH radical scavenging capabilities increase significantly. Both forms impact cancer cell viability differently, with fresh garlic enhancing antioxidant defenses and black garlic modulating energy pathways. These findings confirm that processing methods and biochemical profiles significantly influence garlic's functional properties and potential health benefits, providing insights for targeted therapeutic applications in health and disease management.

Reviewer 2 Report

Comments and Suggestions for Authors The revised version Manuscript ID:molecules-2893577R1:entitled:Comparative Studies of Bioactivities and Chemical Components in Fresh and Black Garlics The revised version of manuscript was improved and accepted for publication. Comments on the Quality of English Language

Minor editing of English language required

Author Response

Authors highly appreciate the Reviewer for the positive evaluation of our manuscript and for the constructive comments and suggestions, which significantly contributed to improving the quality of our work.

Reviewer 3 Report

Comments and Suggestions for Authors

I accept the manuscript after the changes introduced.

Author Response

Authors would like to thank you for your review for the positive evaluation of our revised manuscript.